# Comparing Handcrafted Features and Deep Neural Representations for Domain Generalization in Human Activity Recognition

**DOI:** 10.3390/s22197324

**Published:** 2022-09-27

**Authors:** Nuno Bento, Joana Rebelo, Marília Barandas, André V. Carreiro, Andrea Campagner, Federico Cabitza, Hugo Gamboa

**Affiliations:** 1Associação Fraunhofer Portugal Research, Rua Alfredo Allen 455/461, 4200-135 Porto, Portugal; 2Laboratório de Instrumentação, Engenharia Biomédica e Física da Radiação (LIBPhys–UNL), Departamento de Física, Faculdade de Ciências e Tecnologia (FCT), Universidade Nova de Lisboa, 2829-516 Caparica, Portugal; 3Dipartimento di Informatica, Sistemistica e Comunicazione, Università degli Studi di Milano-Bicocca, 20126 Milan, Italy; 4IRCCS Istituto Ortopedico Galeazzi, 20161 Milan, Italy

**Keywords:** human activity recognition, deep learning, domain generalization, accelerometer

## Abstract

Human Activity Recognition (HAR) has been studied extensively, yet current approaches are not capable of generalizing across different domains (i.e., subjects, devices, or datasets) with acceptable performance. This lack of generalization hinders the applicability of these models in real-world environments. As deep neural networks are becoming increasingly popular in recent work, there is a need for an explicit comparison between handcrafted and deep representations in Out-of-Distribution (OOD) settings. This paper compares both approaches in multiple domains using homogenized public datasets. First, we compare several metrics to validate three different OOD settings. In our main experiments, we then verify that even though deep learning initially outperforms models with handcrafted features, the situation is reversed as the distance from the training distribution increases. These findings support the hypothesis that handcrafted features may generalize better across specific domains.

## 1. Introduction

Human Activity Recognition (HAR) has the objective of automatically recognizing patterns in human movement given sensor-based inputs, namely inertial measurement units (IMUs), currently available in most wearables and smartphones [1]. HAR is an important enabling technology for applications such as remote patient monitoring, locomotor rehabilitation, security, and pedestrian navigation [1].

The IMU itself may contain several sensors, such as accelerometers and gyroscopes, which possess microelectromechanical properties, allowing their capacitance to vary with movement [2]. The accelerometer measures acceleration, while the gyroscope measures angular velocity [3]. Usually, Machine Learning (ML) is applied to enable an association between the signals obtained from these sensors and specific human activities [2]. The typical HAR system comprises the following steps [4]: data acquisition, preprocessing, segmentation, feature extraction, and classification.

Similar to most ML tasks, HAR models perform well when testing on a randomly sampled subset of a carefully acquired dataset (i.e., out-of-sample validation) and struggle in Out-of-Distribution (OOD) settings (i.e., external validation). These settings occur when the source and target domains are different, such as when the models are tested across different datasets or sensor positions [5,6,7].

Deep learning is becoming increasingly popular in HAR applications [8]. While the typical pipeline includes a feature extraction step before training a classifier, deep neural networks automatically learn and extract features through a continuous minimization of a cost function. In principle, a neural network may have millions of learnable parameters, which translates into a large capacity to learn more complex and discriminative features [9]. These models have potential for HAR applications since sensor signals may have many inherent subtleties that may not be recognized by Handcrafted (HC) features. Although a promising approach, significant limitations have been discussed when deep learning models are deployed in real-world environments. Current methods for training deep neural networks may converge to solutions that rely on spurious correlations [10], resulting in models that lack robustness and fail in test domains that are trivial for humans [11].

On the other hand, HC features in this field are well-studied [1,12], more interpretable, and can reach high performance. In  HAR, results with HC features approximate those of deep learning [13,14] even in tasks where the latter thrives, namely when the train and test sets are split by randomly shuffling the data, thus showing similar distributions [15].

Since both methods have advantages and limitations, there is a need for a more detailed comparison between them in various domains. This translates into a need for benchmarks where the similarity between train and test distributions has considerable variability.

As HAR naturally includes many kinds of possible domains, it can be considered an excellent sandbox to study the OOD generalization ability of learning algorithms (Domain Generalization), being previously used for this purpose [16].

This paper compares the performance of learning algorithms based on HC features with deep learning approaches for In-Distribution (ID) and OOD settings. For this comparison, we use five public datasets, homogenized to have the same label space and input shape, so that the models can be easily trained and tested across them. To validate whether the tasks are in fact OOD, several metrics are considered and compared with the purpose of assessing the disparity between train and test sets. To extract HC features, Time Series Feature Extraction Library (TSFEL) [12] was used. We use one-dimensional Convolutional Neural Networks (CNNs) for our deep learning baselines.

In summary, the major contributions of this work are the following:A comparison between different data similarity measures and their relationship to generalization performance.A validation of the hypothesis that models based on HC features can be more robust than deep learning models for several HAR tasks in OOD settings.An empirical demonstration that a hybrid approach between HC features and deep representations can bridge the gap in OOD performance.

## 2. Related Work

Several studies compared classic ML approaches using HC features with deep learning methods. The authors from [13,14,17,18] compare CNNs with models based on support vector machines, multilayer perceptrons, and random forests. In all these studies, deep learning approaches outperformed classic methods. However, in their experiments, data splits were created by randomly shuffling the datasets, so samples from possibly different domains are represented in both the train and test sets with similar data distributions.

In regard to the use of data similarity to quantify the degree of OOD, associated with generalization, this is both an old and important question in the ML literature, as several ML methods implicitly rely on properties related to similarity (e.g., the large margin assumption in SVM learning) to guarantee good generalization performance [19]. The potential relationship between data similarity and the generalization properties of ML models was first investigated from an empirical point of view in [20], where the authors discovered that datasets found to be substantially dissimilar likely stemmed from different distributions. Based on these findings, the authors of [21] demonstrated that information about similarity can be used to understand why a model performs poorly on a validation set, while the same information can be used to understand when and how to successfully perform domain adaptation (see, for example, the recent review [22]). To that end, several metrics for measuring data similarity have been proposed in the literature. Bousquet et al. [20] developed a measure (Data Agreement Criterion, DAC) based on the Kullback–Leibler divergence, which has since become frequently used to assess the similarity of distributions [23]. More recently, Schat et al. [24] suggested a modification to the DAC measure (Data Representativeness Criterion, DRC), and investigated the link between data similarity and generalization performance. Cabitza et al. [25] proposed instead a different approach based on a multivariate statistical testing procedure to obtain a hypothesis test for OOD data, the Degree of Correspondence (DC), and also studied the correlation between DC scores and the generalization of ML models. By contrast, in the Deep Learning literature, approaches based on the use of statistical divergence measures, such as the Wasserstein distance [26] or the Maximum Mean Discrepancy (MMD) [27], have become increasingly popular to design methods for OOD detection. See also, the recent review by Shen et al. [28].

Deep learning approaches have been explored in OOD settings by testing the models on data from unseen domains [4,29,30,31,32]. Gholamiangonabadi et al. [33] verified that the accuracy went from 85.1% when validating using leave-one-subject-out (LOSO) cross-validation to 99.85% when using *k*-fold cross-validation. Bragança et al. [34] had similar results with HC features, reporting an accuracy of 85.37% for LOSO and 98% for *k*-fold. The most important features used by each model differed significantly. They concluded that LOSO would be a better validation method for generalization. Li et al. [4] and Logacjov et al. [30] compared several deep learning models with classic ML pipelines using LOSO validation. As opposed to what was verified in the previous studies involving ID settings, in the context of OOD, classic methods were mostly on par with deep learning approaches, outperforming them in some cases. Still, data acquired from different subjects of the same dataset may not be as diverse as the data encountered by HAR systems in real-world environments since datasets are usually recorded in controlled conditions with similar devices worn in the same positions. In Hoelzemann et al. [7], significant drops in performance were reported when testing on different positions and different datasets, which were then mitigated by the use of transfer learning techniques.

Transfer learning has previously been applied to HAR in cases where feature representations can be used in downstream tasks or across domains [6,35]. These methods leverage information about the target task or domain to approximate the source and target representations [5]. For example, Soleimani et al. [5] used a Generative Adversarial Network (GAN) to adapt the model to each user, outperforming other domain adaptation methods. However, the performance was poor when no transfer learning method was used (see Table 2 of [5]). The same phenomenon can be noticed in [35], where the domain adaptation methods outperformed the baseline model, which did not have access to data from the target domain. These studies illustrate the difficulty of generalizing to different domains, even when using deep learning models.

Gagnon et al. [16] included a HAR dataset in a benchmark to compare domain generalization methods applied to deep neural networks. The results indicate a 9.07% drop in accuracy from 93.35% ID to 84.28% OOD on a dataset where different devices worn in different positions characterize the possible domains. The same study showed that domain generalization techniques [11,36] did not improve results in a significant manner, and that empirical risk minimization (ERM) is still a strong baseline [37].

Boyer et al. [38] compared HC features and deep representations on an ID supervised classification task and on an OOD detection task. They concluded that, while a k-nearest neighbors (KNN) model using deep features as input outperforms the same model using HC features on the ID task, the situation partially reverts for the OOD detection task, where models based on HC features achieve the best results in two out of three datasets. However, the  ID and OOD tasks are not directly comparable, since they are of different kinds and use different evaluation methods.

Trabelsi et al. [39] compared three deep learning approaches and a random forest classifier with handcrafted features as input. Similar to the experiments in our work, the datasets were homogenized by including only common activities and separated the test sets by the user. They concluded that only one of the deep learning approaches outperformed the baseline model with handcrafted features. While they formulated two different domain generalization settings (OOD-U and OOD-MD), the results for each of these settings are not directly comparable since the test sets were combined when reporting the results for the OOD-MD setting.

This paper adds to previous work by explicitly comparing the OOD robustness of HC features and deep representations in four domain generalization settings with different distances between train and test sets.

## 3. Methodology

### 3.1. Datasets

The datasets used in this study include human activity data recorded using smartphones and wearable inertial measurement units (IMUs). Table 1 contains a detailed description of these publicly available datasets.

Several criteria were followed to select the datasets for this study. Only datasets with a sampling rate close to or over 50 Hz were considered, to avoid the need for oversampling. The search was restricted to datasets that included most of the main activities seen in the literature (e.g., walk, sit, stand, run, and ascending/descending stairs). For better compatibility and to avoid large drops in performance caused by having considerably different sensor positions [7], we selected datasets that included overlapping positions with at least one of the other datasets that fulfilled the remaining criteria.

The accelerometer was the selected sensor for this work. The magnitude values were computed as the Euclidean norm of all three axes (*x*, *y*, and *z*), as this quantity is invariant to the orientation of the device and can give information that is more stable across domains. The magnitude signal was used along with the signal from each axis, so that all the information given by the accelerometer was retained. From those four channels, five-second windows were extracted without overlap.

All selected datasets were homogenized [47] so that a model trained on a specific dataset could be directly tested in any other. This procedure included resampling all the recordings to 50 Hz and mapping the different activity labels to a common nomenclature: walking, running, sitting, standing, and stairs. Stair-related labels were joined into a general “stairs” label, as having to distinguish between going up and down the stairs would add unnecessary complexity to the task, since it is hard to infer the direction of vertical displacement without access to a barometer [48]. The RealWorld dataset [46] generated considerably more windows than the other datasets, so one-third of these windows was randomly sampled and used in the experiments. The final distribution of windows and activities per dataset is shown in Table 2. This table contains the percentage of samples (five-second windows) of each activity in a given dataset, as well as the total number of samples and corresponding percentage of each activity and dataset. In this table, it can be seen that, while not being very well balanced, the activities have a substantial amount of samples for all the datasets. On the other hand, even with the effort of reducing samples, the RealWorld and SAD datasets have a larger influence in the experiments, which should not be an issue, since the conditions remain the same for both deep and classic approaches.

### 3.2. Handcrafted Features

To extract HC features, TSFEL [12] was used. This library extracted features directly from the 5-second accelerometer windows generated from each public dataset. To decrease computation time, we removed the features that included individual coefficients, such as Fast Fourier Transform (FFT), empirical Cumulative Distribution Function (eCDF), and histogram values. Nonetheless, the high-level spectral features computed from the FFT were kept. We did not extract wavelet and audio-related features, such as MFCC and LPCC. The total number of features per window was 192.

After the features were computed, samples were split according to each task (see Section 4). Subsequently, features were scaled by subtracting the mean of the train set and dividing by its standard deviation (Z-score normalization). The classifiers used were Logistic Regression (LR) and a Multilayer Perceptron (MLP) with a single hidden layer of 128 neurons and Rectified Linear Unit (ReLU) activation. These classifiers were chosen to enable a fair comparison with deep learning, as they resemble the last layer(s) of a deep neural network, usually responsible for the final prediction after feature learning.

### 3.3. Deep Learning

Convolutional neural networks were the selected deep learning models for this study since they achieved significantly better performance and converged faster when compared with recurrent neural networks (RNN) in preliminary experiments, which was consistent with the literature [49,50]. A scheme of the baseline CNN architectures is presented in Figure 1. We chose three different architectures, which we named CNN-base, CNN-simple, and ResNet. The training process was identical for all the architectures and is explained in Section 4. CNN-simple is a simplified version of the CNN-base with only two convolutional layers and a logistic regression directly applied to the flattened feature maps. ReLU was used as the activation function for the hidden layers of both architectures. The ResNet (Figure 1c) is a residual network inspired by Ferrari et al. [18], with a few modifications. Its convolutional block is represented in Figure 2.

In an attempt to bridge the performance gap between HC features and deep representations, we built a hybrid version of each architecture. There, the  HC features are concatenated with the flattened representations of each model and fed to a fusion layer before entering the final classification layer. The number of hidden units for the fusion layer was 128 on both CNN-simple and CNN-base, increasing to 256 for the ResNet. An illustration of the hybrid version of CNN-base is in Figure 3.

For all these models, the input windows were scaled by Z-score normalization, with mean and standard deviation computed across all the windows of the train set.

### 3.4. Evaluation

To quantify the degree to which a test domain is OOD, different metrics were applied, namely Euclidean distance, Cosine similarity, Wasserstein distance, MMD, and  DC. Each metric was applied to the representations of each model before the classification stage. Regarding the Wasserstein distance [51], the Wasserstein-1 version was used and is given by:(1)$$W_{1}\left(X,Y\right)=\underset{\pi \in \Gamma (X,Y)}{\inf}\int_{R\times R}\left|x-y\right|d\pi \left(x,y\right),$$
where Γ(X,Y) is the set of distributions whose marginals are *X* and *Y* on the first and second factors, respectively. *x* and *y* are samples from each distribution π(x,y) from the set. Intuitively, the distance is given by the optimal cost of moving a distribution until it overlaps with the other. In our experiments, *x* and *y* are the feature representations of subsets of the train and test data, thus $W_{1}$ represents the cost of mapping the distribution of *x* into the distribution of *y* (or vice versa).

Regarding the MMD, this is a kernel-based statistical procedure that aims at determining whether two given datasets come from the same distribution [52]. Given a fixed kernel function k:X×X↦R and two datasets X,Y with sizes |X|=n, |Y|=m, the MMD can be estimated as:(2)$$\mathrm{MMD}\left(X,Y\right)=\frac{1}{n(n-1)}\sum_{i\neq j}k\left(x_{i},x_{j}\right)+\frac{1}{m(m-1)}\sum_{i\neq j}k\left(y_{i},y_{j}\right)-\frac{2}{nm}\sum_{i,j}k\left(x_{i},y_{j}\right)$$

Intuitively, the MMD measures the distance between *X* and *Y* by computing the average similarity in *X* and *Y* separately, and then subtracting the average cross-similarity between the two datasets, where the similarity between two instances is quantified by means of the selected kernel *k*. In this work, a simple linear kernel was selected. Furthermore, as for the Wasserstein distance, *x* and *y* represent the feature representations of subsets of the train and test data. Thus, MMD quantifies the average kernel similarity among instances in *x* and *y*, discounted by the cross-similarity between the two datasets.

The DC, by contrast, is a multivariate hypothesis testing procedure for the hypothesis that two samples of data come from the same distribution: having fixed a representative data sample, the obtained *p*-value, then, can be considered as a measure of how much any other data sample is OOD with respect to the representative one. In particular, scores close to 0 can be interpreted as being most likely OOD (since, assuming the null hypothesis of the two data samples coming from the same distribution, observing a *p*-value close to 0 has low probability). While the DC cannot be defined and computed by means of a closed-form procedure, in [25] a permutation-resampling algorithm (see Algorithm 1) was defined to compute the corresponding *p*-value, based on the selection of a base distance metric.
**Algorithm 1** The algorithm procedure to compute the similarity between the two dataset *T* and *V*, using the Degree of Correspondence (DC). **procedure**
DC(T,V: datasets, *d*: distance, *∂* distance metrics)   $d_{T}=\left\{d\left(t,t^{\prime }\right):t,t^{\prime }\in T\right\}$   For each v∈V, find $t_{v}\in T$, nearest neighbor of *v* in *T*   $T_{|V}=\left\{t\in T:∄v\in Vs.t.t=t_{v}\right\}\cup V$   $d_{T_{|V}}=\left\{d\left(t,t^{\prime }\right):t,t^{\prime }\in T_{|V}\right\}$   $\delta =\partial (d_{T},d_{T_{|V}})$   Compute $\mathrm{DC}=Pr(\delta^{\prime }\geq \delta )$ using a permutation procedure   return DC **end procedure**


The selection of the distance metrics *∂* in Algorithm 1 is important to obtain sensible results for the DC. Intuitively, *∂* should represent the appropriate notion of distance in the instance space of interest. In [53], lacking any appropriate definition of distance in the instance space, the authors suggest the use of a general baseline, e.g., the Euclidean or cosine distance, or robust non-parametric deviation metrics, e.g., MMD or Kolmogorov–Smirnov statistics.

In previous work, model performance has been evaluated using metrics such as accuracy, sensitivity, specificity, precision, recall, and f1-score [1]. As class imbalance is common in most publicly available HAR datasets (see Table 2), f1-score is used as the main performance metric since it is more robust than accuracy in these settings [30]. To be able to compare deep learning models and classic models with HC features, the f1-scores are compared in tasks across different OOD scenarios and including five public HAR datasets.

## 4. Experiments and Results

The main purpose of this paper is to compare the performance of HC features and deep representations in different OOD settings for HAR. A scheme of the full pipeline used for the experiments is presented in Figure 4.

HAR is a classification task that usually involves multiple domains, easily turning into a domain generalization task if the domains are considered when splitting the data. We devise four domain generalization settings, starting with a baseline ID setting where 30% of each dataset is randomly sampled for testing, and three OOD settings: (a) splitting by user within the same dataset, where approximately 30% of the users were assigned to the test set—OOD by user (OOD-U); (b) leaving a dataset out for testing, while including all the others for training—OOD with multiple source datasets (OOD-MD); (c) training on a dataset and leaving another for testing, running all the possible combinations—OOD with a single source dataset (OOD-SD). To obtain a direct comparison, the test set of OOD-U is used as a test set for all the OOD settings. Of the three OOD settings, OOD-U is the one that is expected to be closest to the training distribution since it is drawn from the same dataset, where devices and acquisition conditions are usually similar. It is followed by OOD-MD, since joining all the datasets (except one) for training averages their distributions onto a more general space. Subsequently, as it includes only a single dataset for training, OOD-SD should capture the largest distances between train and test distributions.

In order to validate our hypothesis about the ordering of the distances between the train and test splits on our four settings, different metrics were applied to the feature representations. This experiment has the following objectives: (1) to validate that our three OOD settings are in fact OOD; and (2) to obtain the best metric for our main experiments, which should output values that agree with our ordering hypothesis for both HC features and deep representations. For models based on HC features, metrics were computed directly from the features. In contrast, for deep models, metrics were calculated from the hidden representations of the last layer before classification.

We note that different distance metrics have different scales, therefore, making their interpretation and comparison more difficult. For this reason, we computed distance ratios instead of raw distances, so as to make the values of the different metrics more consistent across tasks. The distance ratios were computed for each task, i.e., setting/dataset combination, using the following equation: (3)$$\mathrm{Distance\_ratio}=\frac{\partial (tr_{1},ts_{1})}{\partial (tr_{2},tr_{3})},$$
where *∂* is a distance metric and $tr_{i}$ and $ts_{i}$ are subsets randomly sampled (with replacement) from the train and test sets, respectively. The sample size is half the minimum of the train and test set lengths. By contrast, for the DC, the raw value without any ratio-based normalization was used, since it is already normalized in the [0,1] range and is able to deal with any data representation directly.

A comparison of the considered metrics based on the TSFEL features is presented in Table 3. It is easy to observe that all the metrics agree with the OOD ordering hypothesis stated above. Indeed, the value of all metrics was higher for the OOD-U, OOD-MD, and OOD-SD (respectively, in this order) than for the ID setting. In particular, it can be seen that DC with Euclidean-based metrics saturates to values close to zero for all three OOD settings, indicating that, by the comments above on the interpretation of this score, the test sets are likely to be OOD.

Table 4 shows a comparison of the considered metrics based on the CNN-base representations. In contrast to the case of TSFEL features, the metrics showed a much lower degree of agreement with the OOD ordering hypothesis. First, it can be noted that only Wasserstein and MMD have values that clearly increase with the expected degree of OOD, being in agreement with the results of the TSFEL representations and, consequently, with our OOD ordering hypothesis. Nonetheless, it can be verified that both metrics had a large degree of variation, with the confidence intervals for the ID, OOD-U, and OOD-MD partially overlapping. In the case of DC Cosine, the score for the OOD datasets was higher than that for the ID one. This seemingly paradoxical behavior may have an intuitive geometric explanation, as it may be a consequence of the transformations that take place during training, which influence the shape of the instance space and possibly make the representations of instances that would otherwise be OOD closer to the training data manifold. In support of this hypothesis, it can be easily observed that most metrics reported a significantly different value for the OOD-SD setting than for the other OOD settings, showing that the training of the deep learning model had an important influence on the natural representation of the data manifold. In this sense, both the Wasserstein and MMD metrics seemed to be more apt at naturally adapting to this change of representation.

Thus, as a consequence of these results, we chose the Wasserstein distance ratio as our main metric to quantifiy the degree of OOD due to the fact that it agrees with our hypothesis when using both TSFEL features and deep representations as input. This metric has also been applied by Soleimani et al. [5] to compute distances between source and target distributions.

Our experiments were run on an NVIDIA (Santa Clara, CA, USA) A16-8C GPU and an AMD (Santa Clara, CA, USA) Epyc 7302 processor with python version 3.8.12 and Visual Studio Code (Microsoft, Redmond, WA, USA) as the development environment. All the learning models were implemented using the PyTorch library [54]. Adam [55] was adopted as the optimizer used for the training process. To reduce bias [16], results were averaged over nine combinations of three different batch sizes (64, 128, and 256) and three learning rates (0.0008, 0.001, and 0.003). To account for class imbalance, the percentage of instances per class in the training set was given to the cross-entropy loss function as class weights.

To make the experiments as agnostic to the training method as possible, the same procedure was used for training the classifiers based on HC features and the deep learning models. Figure 5 shows the training and validation loss over the course of training for a single task. The chosen task was the OOD-U setting on the SAD dataset, an example of a task in which there was a verified occurrence of instability in training. One of the ways to handle this instability is by ending the training process earlier—early stopping [56].

Over all the tasks, most models reached plateaus on validation performance after 30 to 50 epochs, so the training process was limited to 140 epochs to leave a margin for models to converge, but not so much as to fully overfit the data. For validation, we randomly sampled a 10% subset of the training data without replacement. While training, a checkpoint model was saved every time the validation loss achieved its best value since the start of training. Our early stopping method consisted of stopping training if the validation loss did not improve for 30 epochs in a row, which proved helpful in cases where training was not very stable. In these cases, the validation error oscillates, increasing for a certain number of epochs before decreasing again and, on many occasions, achieving a slightly lower error rate than in any of the previous epochs, which can be seen in the loss curves for the CNN models in Figure 5. This resembles the effects of double descent [57]. In our case, one of the causes of such unstable training may be the fact that these datasets are noisy, due to the diversity in users, devices, and positions, among other factors. It may also be a consequence of overparameterization, as the phenomenon was much more pronounced when training CNNs, which have significantly more parameters than our MLP and LR models. Both these potential causes were documented by Nakkiran et al. [57].

The evolution of the f1-score over the Wasserstein distance ratio for the best performing model of each family (CNN-base and TSFEL+LR) is documented in Figure 6. For each combination of model, dataset, and setting, the average and standard deviation of the f1-score were computed over nine different runs with varying learning rates and batch sizes. The CNN-base embeddings were chosen to compute distance ratios for this figure since they contain less outliers when compared to the distance ratios computed from TSFEL representations (see Figure A2). It can be verified that, initially, the CNN model outperforms the model using HC features. However, as the distance between train and test domains increases, the situation is reverted, with the classic approach outperforming the CNN. This suggests that HC features are more robust to the shifts that occur in OOD data. The regression curves reinforce the idea of OOD stability. As expected, there is a negative correlation between f1-score and distance ratio, meaning that performance decreases as the test data becomes more distant from the distribution seen during training. In general, the distance ratios given by the Wasserstein distance appear to agree with the previously stated OOD ordering hypothesis, with OOD-SD being the most OOD of the three settings, followed by OOD-MD and OOD-U, respectively. Still, a few outliers can be seen in the figure. The higher values of standard deviation for the CNN indicate that these models are more susceptible to the choice of hyperparameters, which is reasonable due to the much larger number of trainable parameters. However, it is not always ideal to have such variability, as it indicates that the validation loss has become less correlated with the test loss. In practice, an apparently good model may perform surprisingly well in some settings while failing in situations that would otherwise be trivial to a simple model.

More detailed results are presented in Table 5. For each combination of model and setting, the average and standard deviation of the f1-score were computed over all five datasets. The last column represents the average of the three OOD settings, which gives an idea of the overall generalization performance. The significant overturn from the ID to the OOD settings can be noticed in the table. TSFEL + LR, which had the worst ID f1-score (90.54%), turned out to be the best overall in the OOD regime, with an f1-score of 70% for the average of all three OOD settings. Using an MLP instead of LR slightly decreased the overall OOD performance to 69.55%, while increasing the ID performance to 92.87%, becoming closer to the deep learning results. This phenomenon may be related to an increase in the number of trainable parameters. Including HC features as an auxiliary input to deep models improved both ID and OOD results, with the hybrid version of CNN-base being the deep learning model with the strongest generalization performance (average OOD f1-score of 66.95%). However, this improvement is still insufficient to reach the OOD robustness of models solely based on HC features.

Despite being simpler than the ResNet, the CNN-base model achieves a slightly higher generalization performance. On the other hand, CNN-simple, the simplest deep learning model, did not perform well in OOD tasks. There appears to be an optimal number of parameters, possibly dependent on the architecture, so more studies should be conducted to understand this trade-off.

## 5. Discussion

This work aimed to compare the generalization performance of HC features and deep representations, focusing in particular on generalization in OOD settings.

In the first experiment, several metrics were compared to validate and quantify our OOD settings. For TSFEL representations, all the considered metrics were in agreement with our ordering hypothesis. In particular, the DC was able to clearly identify each of the OOD settings as such. In contrast, for the case of deep representations, there was some disagreement among the considered metrics. Still, the MMD and Wasserstein distance ratios remained in agreement with the adopted hypothesis. They were seen as more robust concerning the change of data representation induced by the deep learning model.

In our experiments involving HAR tasks, despite reaching lower f1-scores in the ID setting, models based on HC features were more robust in OOD settings. This difference in OOD performance supporting higher robustness for HC features may be due to their stability since they are fixed a priori based on domain knowledge, which should be valid across tasks. Conversely, deep features are automatically learned and could thus fail to identify generally helpful features, as there are known inefficiencies in the current methods for training neural networks. These are typically biased toward simple solutions [15] and rely on spurious correlations [10] rather than previous knowledge or causal relations.

In regard to the generalizability of our results to other settings, we note that even though we focused on HAR, with minor adaptations, our experiments and analyses could be replicated in a wide range of fields. For example, similar deep learning models and handcrafted features could be used and compared in fields that depend on sensor data, such as fall detection, predictive maintenance, or physiological signal processing (e.g., EEG, EMG, and ECG). Different deep learning architectures and feature extraction libraries would have to be employed for image or video processing.

Concerning practical purposes, HC features, being more robust, appear to be better suited for real-world HAR systems. However, their reimplementation in mobile or edge devices may be an arduous task. CNNs do not show this limitation, as the representations are encoded in weight matrices and can, in principle, be ported to these devices without significant effort [58]. More studies should, thus, be devoted to exploring this trade-off between increased robustness and reimplementation efforts, possibly considering the application of hybrid approaches (such as the ones also considered in this paper), as well as alternative training techniques for CNNs that attempt to improve robustness.

## 6. Conclusions

This paper hypothesizes that models using HC features generalize better than deep learning models across domains in HAR tasks. Three OOD settings were implemented by testing on unseen users and (single or multi-source) datasets. Five public datasets were homogenized so that they could be combined in different ways to create diverse tasks.

Several metrics were used to quantify the degree of OOD of four domain generalization settings. The DC metric was used to validate our OOD settings. In turn, the Wasserstein distance ratio was chosen as our primary metric for the study since it was able to quantify our three OOD settings in the expected order.

In our main experiments, it was verified that, although deep models have better ID performance, they are outperformed in all three OOD settings by shallow models using features that were computed based on domain knowledge. Furthermore, as the drop in f1-score in OOD settings is less accentuated for classic models, it can be inferred that HC are more robust. Hybrid models achieved intermediate results between deep and classic methods, supporting the idea that HC features can stabilize training, which helps to validate our hypothesis.

Acknowledging the limitation of current deep learning techniques in being robust with respect to OOD settings, as compared to models based on HC features, we believe our work could pave the way for further research on the development of novel training methods for making deep learning models more robust and thus bridge the generalization gap toward new, more trustworthy, gold standards in the field of HAR.

## Figures and Tables

**Figure 1 sensors-22-07324-f001:**
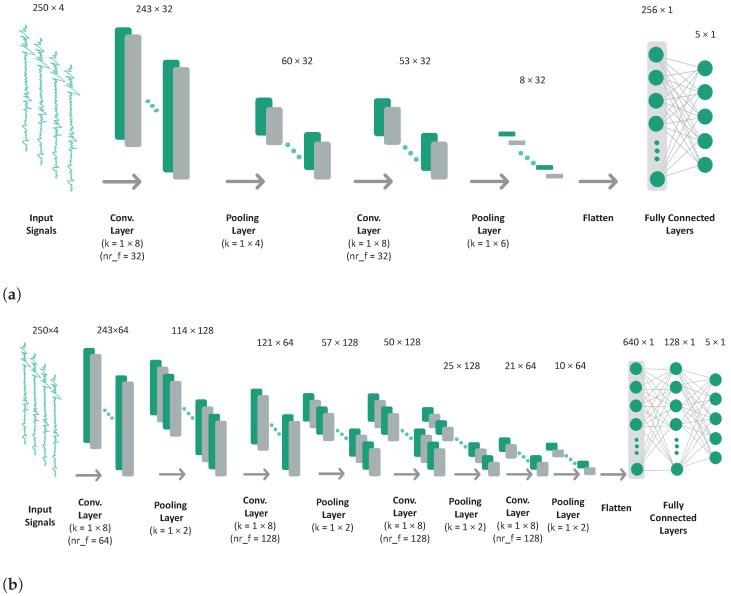
Convolutional neural network architectures. The values above the representation of each feature map indicate their shape (Signal length × Number of channels). Convolutional layers (1D): k = kernel size; nr_f = number of filters; stride = 1; padding = 0. Max pooling layers: k = kernel size; stride = 1; padding = 0. (**a**) CNN-simple Architecture. (**b**) CNN-base Architecture. (**c**) ResNet Architecture. The convolutional block is depicted in Figure 2.

**Figure 2 sensors-22-07324-f002:**
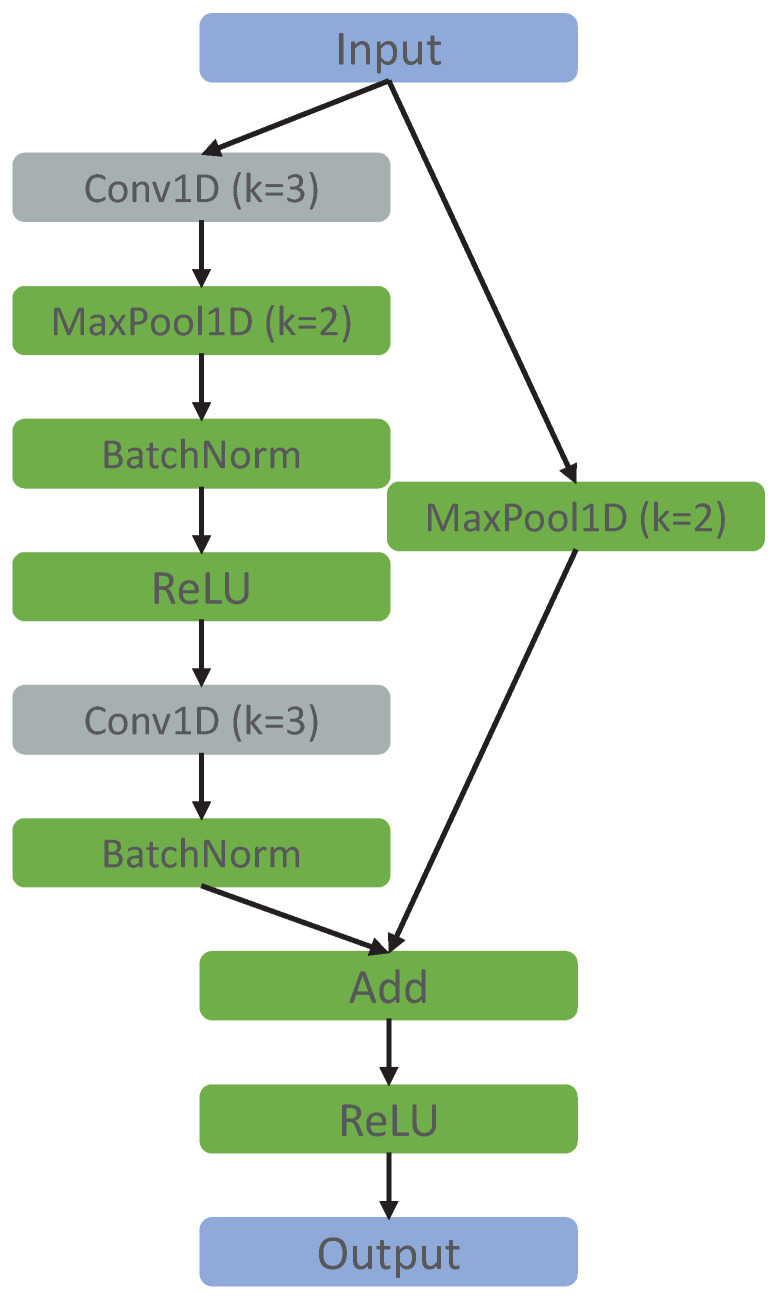
ResNet convolutional block. The letter k stands for “kernel size”.

**Figure 3 sensors-22-07324-f003:**
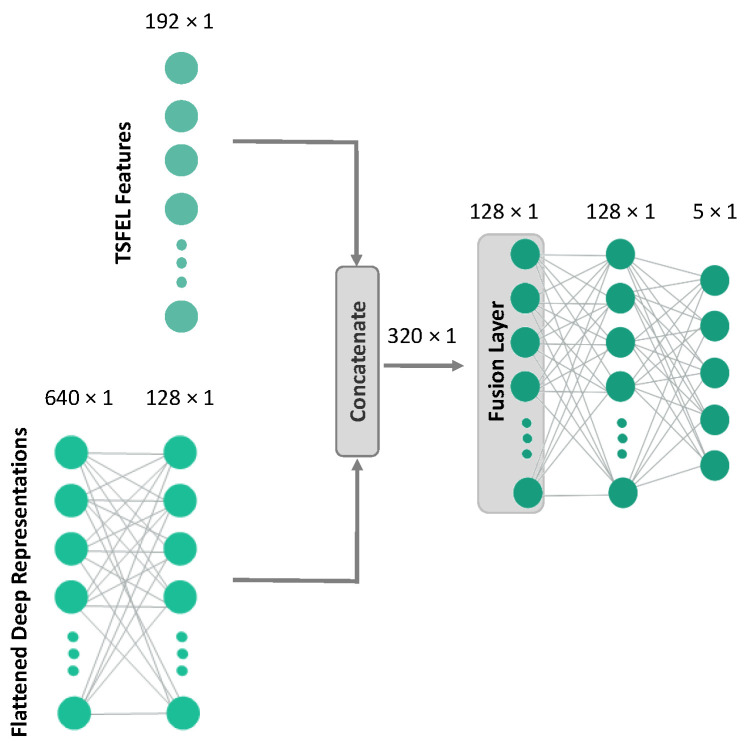
Simplified illustration of the hybrid version of CNN-base (excluding the CNN backbone for ease of visualization).

**Figure 4 sensors-22-07324-f004:**
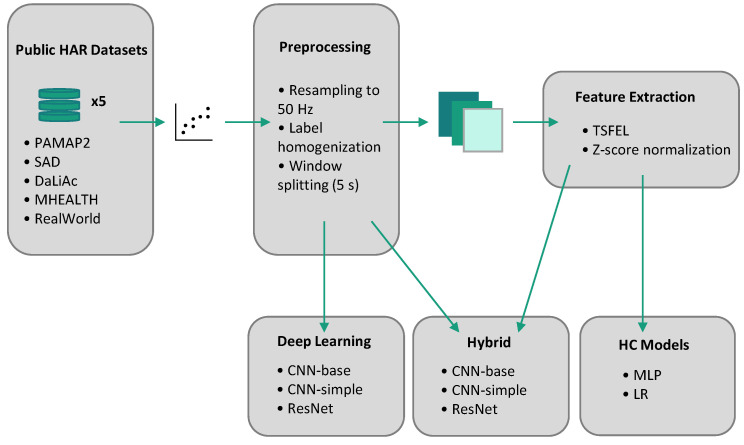
Scheme of the experimental pipeline.

**Figure 5 sensors-22-07324-f005:**
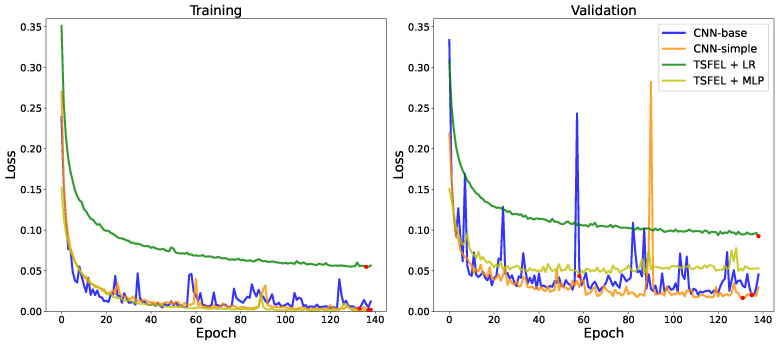
Evolution of loss by epoch on SAD dataset in the OOD-U setting. The red dots indicate the minimum loss of each curve.

**Figure 6 sensors-22-07324-f006:**
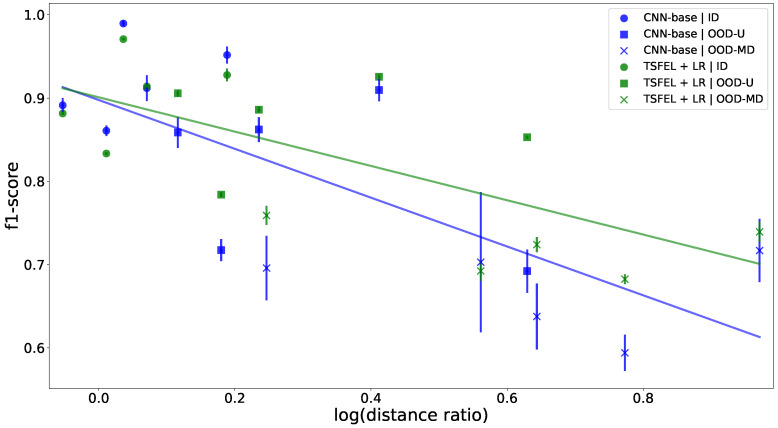
F1-score vs. log(distance ratio). Each marker represents a different task. Distance ratios are based on the CNN-base embeddings. Error bars represent one standard deviation away from the mean. The natural logarithm was applied to the distance ratios to make the regression curves linear.

**Table 1 sensors-22-07324-t001:** Description of the datasets, including activities, positions, devices, and number of subjects.

Dataset	Description	Devices	Source
PAMAP2—Physical Activity Monitoring	9 subjects; 18 physical activities including sitting, lying, standing, walking, ascending stairs, descending stairs and running.	Heart rate monitor (≈9 Hz); 3 inertial measurement units each containing a triaxial accelerometer, a gyroscope and a magnetometer (100 Hz); Positions: wrist, chest and ankle.	[40,41]
Sensors Activity Dataset (SAD)	10 subjects; 7 physical activities: sitting, standing, walking, ascending stairs, descending stairs, running and biking.	5 smartphones containing an accelerometer, a gyroscope and a magnetometer (50 Hz); Positions: jeans pocket, arm, wrist and belt.	[42]
DaLiAc—Daily Life Activities	19 subjects; 13 physical activities including sitting, lying, standing, walking outside, ascending stairs, descending stairs and treadmill running.	4 sensors, each with a triaxial accelerometer and gyroscope (200 Hz); Positions: hip, chest and ankles.	[43]
MHEALTH	10 subjects; 12 physical activities including sitting, lying, standing, walking, climbing/descending stairs, jogging and running.	3 wearable sensors containing an accelerometer, a gyroscope and a magnetometer. One of the sensors also provides 2-lead ECG measurements (50 Hz); Positions: chest, wrist and ankle.	[44,45]
RealWorld (HAR)	15 subjects; 8 physical activities including sitting, lying, standing, walking, ascending stairs, descending stairs and running/jogging.	6 wearable sensors containing accelerometers, gyroscopes and magnetometers (50 Hz). Also includes GPS, light and sound level sensors; Positions: chest, forearm, head, shin, thigh, upper arm, and waist.	[46]

**Table 2 sensors-22-07324-t002:** Distribution of samples and activity labels per dataset. The # symbol represents the number of samples.

	Activity	Datasets (%)					Total	
		PAMAP2	SAD	DaLiAc	MHEALTH	RealWorld	%	#
	Run	10.5	16.9	20.0	33.3	19.1	18.3	7975
	Sit	19.8	16.9	10.6	16.7	17.0	16.3	7102
	Stairs	23.6	32.2	12.3	16.7	30.0	26.3	11,460
	Stand	20.4	16.9	10.6	16.7	16.4	16.2	7047
	Walk	25.7	16.9	46.5	16.7	17.5	22.8	9927
**Total**	**%**	12.7	24.4	15.3	4.96	42.6	-	-
	**#**	5541	10,620	6644	2160	18,546	-	43,511

**Table 3 sensors-22-07324-t003:** Comparison of metrics over all four domain generalization settings based on the TSFEL feature representations. For each setting, values were averaged over every test set. All metrics are ratios except the ones with (*).

Metric	Setting				Avg. OOD
	ID	OOD-U	OOD-MD	OOD-SD	
Wasserstein	1.02±0.04	1.42±0.37	2.27±1.25	3.31±2.39	2.33±0.91
MMD	0.95±0.86	30.47±56.25	800.05±1513.29	1072.20±2619.40	634.24±1008.55
Euclidean	1.00±0.01	1.08±0.11	1.33±0.48	1.53±0.73	1.31±0.29
DC Euclidean *	0.55±0.10	0.05±0.08	0.00±0.00	0.00±0.00	0.02±0.03
Cosine	0.95±0.33	0.85±0.31	0.39±0.52	0.10±0.84	0.45±0.35
DC Cosine *	0.60±0.17	0.32±0.34	0.12±0.16	0.12±0.21	0.19±0.14

**Table 4 sensors-22-07324-t004:** Comparison of metrics over all four domain generalization settings based on the CNN-base representations. For each setting, values were averaged over all the datasets. All metrics are ratios except the ones with (*).

Metric	Setting				Avg. OOD
	ID	OOD-U	OOD-MD	OOD-SD	
Wasserstein	1.06±0.09	1.39±0.27	1.95±0.45	5.71±5.05	3.02±1.69
MMD	1.25±1.00	1.80±0.92	35.23±56.10	245.27±402.93	94.10±135.60
Euclidean	1.00±0.02	1.01±0.05	1.02±0.15	1.12±0.27	1.05±0.11
DC Euclidean *	0.49±0.15	0.51±0.32	0.53±0.45	0.10±0.18	0.38±0.19
Cosine	1.01±0.01	0.98±0.01	0.98±0.03	1.03±0.06	1.00±0.02
DC Cosine *	0.55±0.10	0.92±0.10	0.65±0.43	0.52±0.43	0.70±0.21

**Table 5 sensors-22-07324-t005:** Average f1-score in percentage over all the tasks in a given setting. Values in bold indicate the best performance for each setting.

Model	Setting				Avg. OOD
	ID	OOD-U	OOD-MD	OOD-SD	
CNN-simple	92.09±5.26	79.65±10.75	63.71±3.54	45.21±6.57	62.86±4.36
CNN-base	92.10±5.06	80.79±9.68	66.94±5.19	48.30±5.41	65.34±4.08
ResNet	92.46±4.73	81.16±9.60	67.22±4.89	46.57±4.84	64.98±3.94
CNN-simple hybrid	93.64±4.55	85.13±7.69	66.60±3.31	47.87±2.21	66.53±2.89
CNN-base hybrid	93.48±4.35	85.28±6.64	67.74±3.37	47.84±3.24	66.95±2.71
ResNet hybrid	93.79±4.21	84.71±7.72	67.87±3.40	47.73±2.11	66.77±2.90
TSFEL + MLP	92.87±4.70	87.09±5.35	70.11±3.57	51.45±5.31	69.55±2.78
TSFEL + LR	90.54±5.15	87.08±5.55	71.94±3.19	50.97±3.29	70.00±2.40

## Data Availability

Not applicable.

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
