# Peer review of "Comparing Handcrafted Features and Deep Neural Representations for Domain Generalization in Human Activity Recognition"

_sensors, 2022, doi:10.3390/s22197324_

Round 1

Reviewer 1 Report

The paper is overall written well. The topic is worthy of investigation. I have the following comments to improve the article.

- There are many recent (2022, 2021) papers on HAR, which the authors missed in the literature review. For example, (1) Ohoud Nafea, et al., “Sensor-Based Human Activity Recognition with Spatio-Temporal Deep Learning,” Sensors, vol. 21, no. 6, Article ID: 2141, March 2021. (2) Paola Patricia Ariza-Colpas, et al. "Human Activity Recognition Data Analysis: History, Evolutions, and New Trends," Sensors, 2022, 22, 3401. (3) Ohoud Nafea, et al., “Multi-Sensor Human Activity Recognition using CNN and GRU,” International Journal of Multimedia Information Retrieval, vol. 11, no. 2, pp. 135-147, 2022.  

- I could not understand the description of Tables 3 and 4. Can you please add some sentences using some numbers from these tables?

- A confusion matrix can reveal the performance of the models class-wise. Therefore, you may include a confusion matrix of the models.

- Can you please provide figures of the hybrid versions?

Reviewer 2 Report

1.       The work does not clearly answer one of the ideas in the title, namely the way in which deep neural representations are said to be compared.

2.       In the abstract it is said that the use of neural networks was chosen because this is done in some of the works taken as a reference. It is necessary to clearly explain the scientific and technical properties of neural networks that are useful in the application discussed in the paper.

3.       The abstract talks about an optimal solution, but the paper does not specify which are the optimal indicators that are followed and fulfilled.

4.       In the abstract, it is said that comparisons are made from several fields, but the paper does not clearly show how the method is applied in each of these fields.

5.       It is necessary to use in the bibliography more works with approaches closer to the one in the present work.

6.       A more detailed description of the data presented in Tab. 2.

7.       Fig. 1 is unclear. It is necessary to increase it.

8.       It is necessary that the information in paragraph 3.3 be accompanied by several graphic schemes.

9.       It is necessary to use a graphic scheme from which it can be deduced what the quantities are from relations (1) and (2) and to describe their method of acquisition in the process.

10.   Fig. 2 is totally unclear. It must be redone. But also explained in detail.

11.   The work does not give any information about the development of neural networks, which makes the results obtained unreliable. In the development of neural networks there are two phases: the training or learning phase and the use phase in the application. Give specific details of the networks used, so that readers can reproduce the model.

12.   There is no scheme of the equipment, programs, acquisition, processing and calculation algorithms used.

13.   The work is analyzed for a magazine in the field of sensors. As such, it is necessary to detail how the sensors are used to solve the problem.

14.   Give details about how relation (3) is used in the application, what are the physical quantities used and how their values are acquired.

15.   It is necessary to explain the scenario used after which the results in Fig. 3.

16.   It is necessary to explain the variables from Table 5.

17.   Insert a discussion chapter of the results.

18.   In my opinion, the conclusion chapter does not respond to the theme proposed in the title of the work, nor to the issues highlighted in the abstract.

Round 2

Reviewer 1 Report

The authors addressed most of the comments. 

Author Response

Thank you very much. Your comments certainly improved the quality of our paper.

Reviewer 2 Report

The authors did not answer the questions properly: Please give more consistent answers.

Authors must respond in much more detail to each of the observations made, such as, for example, to the observations: 1, 2, 3, 4, 12, 13, 15, 16.

Author Response

Thank you for your effort in the review. Please see the attachment.
